# Equitable Stable Matchings in Quadratic Time

**Nikolaos Tziavelis**
Northeastern University

**Ioannis Giannakopoulos**
NTU Athens

**Katerina Doka**
NTU Athens

**Nectarios Koziris**
NTU Athens

**Panagiotis Karras**
Aarhus University

## Abstract

Can we reach a stable matching that achieves *high equity* among the two sides of a market in *quadratic time*? The *Deferred Acceptance* (DA) algorithm finds a stable matching that is biased in favor of one side; optimizing apt equity measures is strongly NP-hard. A proposed approximation algorithm offers a guarantee only with respect to the DA solutions. Recent work introduced *Deferred Acceptance with Compensation Chains* (DACC), a class of algorithms that can reach any stable matching in $\mathcal{O}(n^4)$ time, but did not propose a way to achieve good equity. In this paper, we propose an alternative that is computationally simpler and achieves high equity too. We introduce *Monotonic Deferred Acceptance* (MDA), a class of algorithms that progresses monotonically towards a stable matching; we couple MDA with a mechanism we call *Strongly Deferred Acceptance* (SDA), to build an algorithm that reaches an equitable stable matching in quadratic time; we amend this algorithm with a few low-cost local search steps to build *Deferred Local Search* (DLS), which, as we demonstrate experimentally, outperforms previous solutions in terms of equity measures and matches the most efficient ones in runtime.

## 1 Introduction

A *matching* process on a two-sided market can determine who gets which job [40, 31], school place [44], or spouse. Gale and Shapley [16] proposed[1] a model, in which each agent (e.g., woman or man) ranks members of the other set by strict order of preference; then agents on the one side issue proposals (i.e., offers) to those on the other side by that order; recipients hold the best proposal they have received, without commitment, until nobody wishes to propose. This $\mathcal{O}(n^2)$ algorithm, called the *Deferred Acceptance* (DA) algorithm in contradistinction to *immediate acceptance* [41], leads to a *stable* solution; that is, no pair of agents would rather be matched with each other than with their assigned partners. The DA algorithm has had a profound influence on market design and stands at the basis of a number of centralized labor market clearinghouses around the world, allowing failed markets to be reorganized [41]. Roth and Shapley shared the Nobel Memorial Prize in Economic Sciences for their work in "the theory of stable allocations and the practice of market design", reflecting also Roth's application of these results to real-world markets [32].

The problem instance size may be large. In China, over 10 million students apply for admission to higher education institutions annually through a centralized process [32]. Similar centralized schemes, in which students apply for education programs and are ranked according to their scores [8], occur in Germany [9], Greece, Hungary [7], Ireland, Spain [37], Turkey [6, 8, 4], and several school districts in the USA [1, 3, 2]. Apart from the instance size, the set of possible stable matchings is large in real-world markets [20], and exponentially growing in the worst case [29, 30]. Still, the DA algorithm returns a solution *optimal* for proposers, as each proposer gets the best match possible in any stable

matching, and, in reverse, *pessimal* for recipients [16, 34]; thus, it finds either the *man-optimal* or the *woman-optimal* stable matching. Yet many real-world markets require stable matchings that are fair to both sides [19, 43, 15]. For example, in a health care market, each surgeon may have preferences for which anesthetist to work with, and vice versa; an impartial allocation that eschews any favoritism would arguably lead to a sense of fairness and better performance [39]. There is then a practical need to find stable matchings that do not favor any side. Unfortunately, minimizing apt measures of *equity* or *balance* between the two sides is NP-hard [26, 15]. An approximation algorithm [25] provides a guarantee only with respect to the *biased* DA solutions. Thus, there is a need for efficient and effective algorithms that produce equitable stable matchings [21, 42].

In this paper, we provide the first, to our knowledge, quadratic-time algorithms that reach stable matchings of good equity measures. We first introduce a class of algorithms called *Monotonic Deferred Acceptance* (MDA), which exploit the growth of a monotonic state function; then, we introduce a new proposal mechanism, *Strongly Deferred Acceptance* (SDA), by which an agent cannot be in a pair and issue proposals at the same time. We devise an algorithm utilizing MDA and SDA, POWERBALANCE, that terminates in $\mathcal{O}(n^2)$ time, and enhance it with a few selective low-cost local search steps to produce even more equitable solutions. We call the full operation *Deferred Local Search* (DLS). Our experimental study with simulated markets shows that DLS outperforms the state of the art in equity measures and matches the most efficient heuristics in runtime.

## 2 Background and Related Work

An instance $I$ of the *stable marriage problem* (SMP) comprises of a set $\mathcal{W} = \{w_1, w_2, \ldots, w_n\}$ of $n$ women and a set $\mathcal{M} = \{m_1, m_2, \ldots, m_n\}$ of $n$ men, and for each person (or *agent*) a *preference list*, i.e., a total order of the members of the opposite side from most to least preferable. Let $\ell_q$ be the preference list of agent $q$; $\ell_q[k] = p$ means that $q$ ranks $p$ as its $k^{\text{th}}$ preference, with $k = 0$ denoting the highest preference; we also write $pr_q(p) = k$. If a woman $w$ prefers $m_1$ to $m_2$, i.e., $pr_w(m_1) < pr_w(m_2)$, we denote that as $m_1 \succ_w m_2$; likewise for men's preferences. A (perfect) *matching* $\mu$ on $I$ is a set of $n$ disjoint man-woman pairs. If a woman $w$ and a man $m$ are matched in $\mu$, we write $\mu(w) = m$ and $\mu(m) = w$. A woman $w$ and a man $m$ form a *blocking pair* for $\mu$ when: (i) $\mu(m) \neq w$; (ii) $w \succ_m \mu(m)$; and (iii) $m \succ_w \mu(w)$. A matching $\mu$ is *stable* if no blocking pair exists for $\mu$, otherwise it is *unstable*. The SMP calls for finding a stable matching.

**The Deferred Acceptance Algorithm**  In the Deferred Acceptance (DA) algorithm [16], each man $m$ starts out from his first preference, with an index $\kappa_m = 0$, and proposes to the woman at entry $\ell_m[\kappa_m]$, increasing $\kappa_m$ in each iteration, as long as he remains unmatched. A woman $w$ accepts a proposal from a man $m$ to form pair $(w, m)$ if she is single or $m$ is more preferable to her than the current fiancé, $\mu(w)$. We express this acceptance condition by the following Boolean predicate:

$$\texttt{accept}(w, m) = \texttt{single}(w) \ \lor \ m \succ_w \mu(w), \tag{1}$$

where $\mu$ is the matching created so far. If the proposal is rejected, $m$ moves to preference $\kappa_m + 1$. The DA algorithm reaches a stable matching in $\mathcal{O}(n^2)$ steps [19]; the number of pairs never decreases: when a woman breaks one pair and creates another, her preference for her fiancé may only improve; contrariwise, a man's preference for his fiancée may only worsen.

**Breakmarriage and Rotations**  The DA algorithm is biased: it returns, out of a set of stable matchings that may grow exponentially in the worst case [22], one that is most preferable to each proposing agent and least preferable to each recipient agent [16, 34]. For example, if men's first preferences do not conflict, each man may obtain his first choice, regardless of how satisfactory that is to women. The complete set of stable matchings for a problem instance forms a *distributive lattice* under a natural *dominance* relation[2], in which the unique maximum and minimum elements are the two gender-optimal matchings [29, 30]. This lattice can be traversed through *breakmarriage* operations [34]: starting out from a stable matching $\mu$, we break a pair $(m, w)$; then man $m$ proceeds as per the DA algorithm, initiating a sequence of proposals that terminates either with a man being rejected by all women (a dead-end) or to a new stable matching $\mu'$. During this operation, there is exactly one single man at any time, who makes the next proposal. The resulting stable matching is *dominated* by the initial one, in the sense that all men who changed partners are worse off.

A breakmarriage operation [34] corresponds to one or more *rotations*, i.e., minimal operations whereby a cyclically ordered sequence of pairs exchange partners, transforming one stable matching to another [22]. A *precedence relation* defines a *partial order* by which rotations can be performed, the *rotation poset*. Each stable matching corresponds to a closed subset of the rotation poset [22]; applying this subset of rotations from one lattice end, in any valid order, results to the same stable matching. All rotations are found in $\mathcal{O}(n^2)$ time via breakmarriage operations [18, 19].

**Defining Fairness**  The bias of the DA algorithm calls for solutions that optimize some measure of fairness. Knuth [29, 30] describes, with credit to Selkow, an $\mathcal{O}(n^4)$ algorithm to find a stable matching $\mu$ that minimizes the lowest preference assigned to any agent, or *regret cost* $r(\mu)$; others proposed an $\mathcal{O}(n^2)$ algorithm [18] and another $\mathcal{O}(n^4)$ algorithm to the same effect [39]. Still, a minimum-regret matching may coincide with one of the DA outputs, even when there are many stable matchings [21]. To consider the big picture, we define two quantities: the sums of women's and men's preferences for their matches in a stable matching $\mu$: $\wp_1 = \sum_{(m,w) \in \mu} pr_m(w)$, $\wp_2 = \sum_{(m,w) \in \mu} pr_w(m)$.

Given a pair $(m, w) \in \mu$, $m$ envies the partners of all women $w'$ such that $w' \succ_m w$; the *egalitarian cost* [19] is a measure of fairness that counts the number of *envy situations* in the market [38]. $Eg(\mu) = \wp_1 + \wp_2$. A stable matching of minimum egalitarian cost is found in $\mathcal{O}(n^3)$ [23, 14]. Still, by such a matching, one side may fare much better than the other. The *sex equality cost* [19] measures the gap between the two sides' sums of preferences for their matches: $SEq(\mu) = |\wp_1 - \wp_2|$.

Still, sex-equality may compromise overall happiness: by this measure, a stable matching in which the two sides are closer to each other *is preferred* over another matching in which *both* sides fare better, but at an increased gap. The *balance cost* [15] provides an alternative: $Bal(\mu) = \max\{\wp_1, \wp_2\}$, minimizing the unhappiness of the most unhappy side [33]. Our goal is to find stable matchings of low sex equality and balance cost. Unfortunately, minimizing the sex equality cost is strongly NP-hard [26, 33]. Iwama et al. [25] gave an $\mathcal{O}\left(n^{3+\frac{1}{\epsilon}}\right)$ algorithm, which for some fixed $\varepsilon > 0$, returns a matching $\mu$ such that $SEq(\mu) \leq \varepsilon\Delta$, where $\Delta$ is the least sex-equality cost among the two DA outputs, or reports that no such matching exists. We revisit this algorithm in our experimental study. Minimizing balance is also NP-hard [15]. Manlove [32] constructs an instance, credited to McDermid, in which no balanced stable matching is a sex-equal stable matching, and vice versa.

**DA-Extending Procedures**  Past research [13, 35, 17, 11] has proposed procedures that aim to find a fair stable marriage by extending the DA algorithm; they allow agents on both sides to issue proposals, one after another, each agent following the order of its preference list. At any time, $\kappa_a$ denotes the position on the preference list of agent $a$ where $a$ issues a proposal when its turn comes; $\kappa_a$ increases with every rejection. When $a$ accepts a proposal from $b$, such that $b \succ_a \ell_a[\kappa_a]$, then it sets $\kappa_a = pr_a(b)$, i.e., it upgrades $\kappa_a$ to the position of $b$ in its preference list, so that it resumes proposals in case of a divorce. Yet an agent $a$ may not *skip forward* positions in its preference list. We call this class of algorithms [13, 35, 17, 11] **DA-extending procedures**; all DA-extending procedures arrive at a stable matching $\mu$ *iff* each agent $a$ is in a couple with its preference at $\kappa_a$, i.e., $\mu(a) = \ell_a[\kappa_a]$. However, they may enter endless loops. Dworczak [11] suggests a variant, *Deferred Acceptance with Compensation Chains* (DACC), that *immediately compensates* any agent $a$ abandoned by a partner that had proposed to $a$, letting $a$ issue proposals until it finds a new partner. Dworczak [11] does not prove termination in the case in which two divorcees need to be compensated in the same round of the algorithm, and gives no polynomial runtime bound for DACC; after communication with the author, we have ascertained that DACC terminates in $\mathcal{O}(n^4)$ [12]. Still, there is no suggestion in [11] on how to generate operations that quickly converge to a solution achieving high fairness. The main idea of DACC is reminiscent of EROM [39, 27], a regret-minimizing $\mathcal{O}(n^4)$ procedure that lets all agents propose with progressive receptiveness: in round $k$, only preferences ranked up to $k$ may be proposed to and accepted. EROM compensates *every* agent abandoned by its partner; at its final stage, when $k = n$, it enacts compensation chains that go on until a single agent accepts a proposal. Yet, contrary to DACC, EROM only accesses a regret-minimizing sublattice of the stable marriage lattice. In this restrictive nature, EROM is akin to LOTTO [5], a *random serial dictatorship* mechanism that reduces the space of attainable stable matchings in favor of a randomly chosen agent in each iteration.

In another direction,  a local search algorithm, BILS [45], starts out from the two DA solutions and bidirectionally traverses the lattice of stable marriages via breakmarriage operations, guided by a cost measure. When the two operations meet each other in terms of cost, it outputs the one of best cost.

# 3 Enforcing Monotonicity

We aim to provide a procedurally fair [28] DA-extending procedure that converges to a stable matching of high equity in quadratic time. We first introduce some basic concepts.

**Definition 3.1** (Proposal index). *Given a set of agents $\mathcal{A} = \{a_i\}$ in a two-sided market, the* proposal index $\kappa_a$ *of an agent $a$ is the index of the position in $a$'s preference list, such that $a$ intends to make its next proposal (offer) to the agent $\ell_a[\kappa_a]$ (or to none, if $\kappa_a = n$); $\kappa_a$ advances to the next position with each rejection, yet* backtracks *to the position $pr_a(b)$ of an agent $b$ who proposes to $a$, if $b \succ_a \ell_a[\kappa_a]$.*

**Definition 3.2** (State). *Given a set of agents $\mathcal{A} = \{a_i\}$ in a two-sided market the* state *of $\mathcal{A}$ at a given time is the set $\{\kappa_{a_i}\}$, where $\kappa_{a_i}$ is $a_i$'s current proposal index value.*

**Definition 3.3** (Frontier index). *Given a set of agents $\mathcal{A} = \{a_i\}$ in a two-sided market the* frontier index $\lambda_a$ *of each agent $a$ is the largest value that $a$'s proposal index $\kappa_a$ has assumed so far, i.e., the farthest position in $a$'s preference list to which $a$ has ever made an offer.*

**Definition 3.4** (Idle agent). *An agent $a$ is* idle *when it has proposed to all its preferences up to its current match or the end of its preference list, i.e., $\kappa_a = pr_a(\mu(a))$ or $\kappa_a = n$.*

**Definition 3.5** (Idle couple). *A couple $\{a, b\} \in \mu$ is* idle *when both of its members are idle, i.e., $\kappa_a = pr_a(b)$ and $\kappa_b = pr_b(a)$, hence $a = \ell_b[\kappa_b]$ and $b = \ell_a[\kappa_a]$; in other words, $a$ and $b$ have both proposed to each other and none of them is still making offers to other options.*

**Monotonic Events**  A procedure of proposals issued by both sides that does not terminate must eventually bring $\mathcal{A}$ back to a state where it has already been. In reverse, as long as a procedure brings $\mathcal{A}$ to states where it has *never* been before, it is not in a loop. Thus, if we reach a state never encountered before, then an algorithm is not looping. We can determine that we reach a state never encountered before when a *monotonically non-decreasing* function of state grows to a value never reached before. By enforcing the growth of such functions, we ensure that the algorithm in question does not loop. We call an event of growth of such a function a *monotonic event*.

**Definition 3.6** (Monotonic event). *Given a set of agents $\mathcal{A} = \{a_i\}$ in a two-sided market and an algorithm operating on it, a* monotonic event *is the increase of a function that is monotonically non-decreasing during the algorithm's operation and upper-bounded by a maximum value.*

We now define two such functions. Each frontier index $\lambda_a$ is *monotonically non-decreasing*, as by definition it cannot be decreased during an algorithm's operation, and is upper-bounded by $n$. Thus:

**Corollary 3.1.** *The increase of a frontier index $\lambda_a$ is a monotonic event.*

The number of idle couples $\mathcal{C}$ is also *monotonically non-decreasing*: if such a couple is broken by one partner $a$, then $a$ accepts a proposal from a more preferable option $b$, and thereby remains idle with $\mu(a) = b = \ell_a[\kappa_a]$, while the proposing agent $b$ becomes idle, as it has just proposed to $a = \ell_b[\kappa_b]$.

**Corollary 3.2.** *The increase of $\mathcal{C} = |\{\{a, b\} \in M | a = \ell_b[\kappa_b] \wedge b = \ell_a[\kappa_a]\}|$ is a monotonic event.*

We call the class of algorithms that enforce monotonic events *Monotonic Deferred Acceptance* (MDA). The following theorem defines an example of an MDA procedure.

**Theorem 3.1.** *Assume an algorithm operates on a set of agents $\mathcal{A} = \{a_i\}$ in a two-sided market, starting from any state. Then continuous proposals by agents on the same side will lead, in at most $\mathcal{O}(n^2)$ steps, to one of the following events: (i) a frontier index $\lambda_a$ increases, or (ii) a new, additional idle couple is formed, hence $\mathcal{C}$ increases, or (iii) all agents on the proposing side become* idle.

*Proof.* As proposing-side agents do not receive offers, none of them rises to a more preferable position in its preference list. Thus, each proposing-side agent $a$ increases $\kappa_a$. Eventually, one of them may reach and exceed $\lambda_a$, a monotonic event. Alternatively, an agent $a$, may issue a proposal and form a new idle couple (either by proposing to its current match $\mu(a)$ or to a single agent) before it reaches $\lambda_a$, also a monotonic event. If no such event occurs, then each proposing agent $a$ either (i) is already idle, or (ii) has a proposal accepted at $\kappa_a \leq \lambda_a$ and becomes idle *without* forming an additional idle couple, or, (iii) has $\lambda_a = n$ and $\kappa_a$ reaches that terminal position, hence again $a$ becomes idle. Therefore, eventually either a monotonic event occurs, or all agents on the proposing side are rendered idle; that happens in at most $\mathcal{O}(n^2)$ steps, the amount of all possible proposals. $\square$

Cases (i) and (ii) in Theorem 3.1 constitute monotonic events. If such an event occurs, the algorithm is not in a loop; we can then switch from the one side, $A$, to the other side, $B$, so as to give to agents on both sides the opportunity to receive and issue proposals, and insist on side $B$ until a monotonic even occurs; the sooner a monotonic event occurs and we switch side, the more evenly we treat the two sides. However, no monotonic event occurs in Case (iii), when all agents on side $A$ are rendered idle. It is tempting to think that, with all agents on side $A$ already idle, the termination of the algorithm is imminent, after a few proposals from side $B$. Unfortunately, this is not the case, as there may exist a couple $\{a, b\}$ with an idle agent $a$ on side $A$ and a non-idle partner $b$ on side $B$, i.e., with $\ell_a[\kappa_a] = b$ but $\ell_b[\kappa_b] \succ_b a$; after switching to side $B$, $b$ may propose to others on side $A$ and hence abandon $a$; thereby, $a$ is rendered non-idle, and hence we will still need to return to proposing with side $A$. In other words, the allowance for couples in which one partner is non-idle[3] renders the termination of the algorithm problematic and calls for measures like those in [11], which incur a high computational overhead. In the following, we introduce our proposal that overcomes this problem.

# 4 Strongly Deferred Acceptance

Since termination is rendered problematic by couples that contain a non-idle agent, we reason that we should disallow the creation of such couples in the first place; in other words, every couple should be an *idle couple*. By that precaution, once all agents on one side, $A$, are rendered idle, no agent $a$ on side $A$ can be abandoned by its partner: if such partner $b$ exists, it is necessarily idle, and every agent on side $A$ that could propose to $b$ is idle too. Then, as we will show, after all agents on side $A$ are rendered idle, the algorithm can securely terminate by letting agents on side $B$ propose. Yet, to disallow the creation of couples with a non-idle agent, we should modify the proposal acceptance criterion in Equation (1), employed by the DA and DA-extending algorithms [16, 17, 11]. By this criterion, as discussed in Section 2, an unmatched agent $a$ accepts a proposal from any agent $b$ on the other side; thereafter, it may continue issuing proposals of its own, as long as $\kappa_a < pr_a(b)$, i.e., $\ell_a[\kappa_a] \succ_a b$. We propose a simpler acceptance criterion that eschews this duplicity: an agent $q$ accepts a proposal from another agent $p$ *if and only if* $p$ is preferable to $q$ over its next proposal target:

$$\mathtt{accept}(q, p) = p \succ_q \ell_q[\kappa_q] \tag{2}$$

In case of acceptance, $q$ sets $\kappa_q = pr_q(p)$, otherwise $p$ moves on to preference $\kappa_p + 1$. We call this mechanism *Strongly Deferred Acceptance* (SDA).

**Properties**   We now study the capacity of an SDA procedure using an arbitrary order of proposals to terminate to stable solutions from a given starting state, i.e., its *stability* and *reachability* properties.

**Definition 4.1.** *An SDA proposal procedure* terminates *when it brings all agents to an* idle *state.*

**Definition 4.2.** *Given a stable matching $\mu$, we characterize the position of agent $p$ with respect to $\mu$ in terms of its $\kappa_p$ as follows: (i) if $\kappa_p < pr_p(\mu(p))$, $p$ is $\mu$-overrated, i.e., proposing above its match in $\mu$; (ii) if $\kappa_p = pr_p(\mu(p))$, $p$ is $\mu$-pivotal, i.e., ready to propose to its assignee in $\mu$; (iii) if $\kappa_p > pr_p(\mu(p))$, $p$ is $\mu$-underrated, i.e., has been already rejected by its match in $\mu$.*

**Lemma 4.1.** *Given two agents $p$ and $q$, during the operation of an algorithm issuing proposals by both sides, starting with $\kappa_a = 0 \quad \forall a \in \mathcal{A}$, there can be no state in which $\kappa_p > pr_p(q)$ and $\kappa_q > pr_q(p)$.*

*Proof.* Assume $p$ and $q$ find themselves in such a position. Then one of the two, say $p$, must have exceeded its preference for the other, $q$, while $q$ was already in such a position. Then $q$ must have rejected a proposal from $p$ while $\kappa_q > pr_q[p]$, i.e., $p \succ_q \ell_q[\kappa_q]$; that cannot happen: $q$ should have accepted the proposal from $p$, since $p \succ_q \ell_q[\kappa_q]$. $\qquad\square$

**Theorem 4.1** (Stability). *When a procedure by SDA terminates, the outcome is a stable matching.*

*Proof.* Suppose the resulting matching contains a blocking pair $(x, y)$, i.e., $\kappa_x > pr_x[y]$ and $\kappa_y > pr_y[x]$; that is a violation of Lemma 4.1. Hence the theorem follows. $\qquad\square$

**Lemma 4.2.** *By SDA, an $\mu$-overrated agent $p$ may only form a couple with an $\mu$-underrated agent.*

*Proof.* Let $p$ be an $\mu$-overrated agent that forms a couple with $q$. If $q$ is $\mu$-overrated, then $(p, q)$ would be a blocking pair in $\mu$, hence $\mu$ would not be stable. If $q$ is pivotal, then $p = \mu(q)$, hence $q = \mu(p)$, thus $p$ cannot be overrated. Hence the lemma follows. □

**Theorem 4.2** (Universality). *Starting from the state $\{\kappa_i = 0, \forall a_i \in \mathcal{A}\}$, any stable matching $\mu$ is reachable by SDA proposals.*

*Proof.* Let $\mu$ be any stable matching. Initially, all agents are $\mu$-overrated. Let each agent $a$ propose to all preferences up to $\kappa_a = pr_a(\mu(a))$; by Lemma 4.2, these proposals cannot be accepted, as they would form pairs among $\mu$-overrated agents; then all agents are $\mu$-pivotal, hence produce $\mu$. □

**Exploiting SDA**  The following theorem shows how we can achieve termination by SDA.

**Theorem 4.3.** *Assume an algorithm operates on a set of agents $\mathcal{A} = \{a_i\}$ in a two-sided market under SDA, starting from a state in which all agents on one side, $A$, are idle. Then continuous proposals issued from the other side, $B$, lead, in $\mathcal{O}(n^2)$ steps, to a stable matching $\mu$.*

*Proof.* By Theorem 4.1, to show that the outcome is a stable matching, it suffices to show the procedure terminates with all agents rendered idle. Each agent $a$ on side $A$ remains idle during proposals from side $B$, since it has either reached $\kappa_a = n$ as a single, or is matched to an idle partner $b$, and remains idle in case it accepts a proposal from another agent $b'$ on side $B$. As only agents on side $B$ propose, eventually they all are rendered idle too. The process requires proposals at most equal to the the length of the preference lists of side $B$, hence $\mathcal{O}(n^2)$ proposals. □

We call the total two-round process COMPROMISE. The critical point is that, by SDA, once all agents on one side are idle, none of them can lose its partner, who is also idle. Contrariwise, by DA, an idle agent may be abandoned by a non-idle partner, hence termination does not come forth in two rounds.

We now develop an algorithm that terminates efficiently and caters to fairness too. We propose an initial phase in which the two sides both propose in turns, followed by a COMPROMISE phase. If COMPROMISE is applied on the initial state with side $A$ proposing first, it produces the $B$-side-optimal stable marriage. This may sound counterintuitive, given that the DA algorithm obtains a proposer-optimal matching [16]. Yet, in DA, the other side *never* proposes. When both sides propose in turns, the advantage is with the one that *receives* proposals first. Thus, it is fair to assign the role of proposers in each round to the side deemed to be better off, as measured by their $\kappa$ index values.

---

**Algorithm 1** PowerBalance

**Input:** $\mathcal{A} = \mathcal{M} \cup \mathcal{W}$ (men and women), $limit$, $cost$
**Output:** stable matching $\mu$
    $n = |\mathcal{M}| = |\mathcal{W}|$; $\mu = \emptyset$; $Rounds = 0$
    **for all** $x \in \mathcal{A}$ **do** $\kappa_x = 0$
    **while** $(|\mu| < n)$ **do**
        $Rounds$++; $P = $ STRONGSIDE$(\mathcal{M}, \mathcal{W})$
        **for all** $p \in P$ **do** PROPOSE$(p,\mu)$
        **if** $(Rounds > limit)$ **then**                        ▷ Enforce termination after $limit$ rounds
           $\mu_1 = $ COMPROMISE$(\mathcal{M}, \mu)$; $\mu_2 = $ COMPROMISE$(\mathcal{W}, \mu)$
           **if** $(cost(\mu_1) \leq cost(\mu_2))$ **then** $\mu = \mu_1$ **else** $\mu = \mu_2$
    **return** $\mu$
    **function** STRONGSIDE$(\mathcal{M}, \mathcal{W})$
        **if** $(\sum_{m \in \mathcal{M}} \kappa_m \leq \sum_{w \in \mathcal{W}} \kappa_w)$ **then return** $\mathcal{M}$ **else return** $\mathcal{W}$
    **function** COMPROMISE$(\mathcal{C}, \mu)$                            ▷ side $\mathcal{C}$, matching $\mu$
        **if** $(\mathcal{C} == \mathcal{M})$ **then** $\mathcal{F} = \mathcal{W}$ **else** $\mathcal{F} = \mathcal{M}$
        **while** $(\exists x \in \mathcal{C} : \mu(x) = \emptyset \wedge \kappa_x < n)$ **do**           ▷ Render side $\mathcal{C}$ idle
           **for all** $x \in \mathcal{C}$ **do** PROPOSE$(x,\mu)$
        **while** $(\exists x \in \mathcal{F} : \mu(x) = \emptyset \wedge \kappa_x < n)$ **do**       ▷ Side $\mathcal{F}$ completes the matching
           **for all** $x \in \mathcal{F}$ **do** PROPOSE$(x,\mu)$
        **return** $\mu$
    **procedure** PROPOSE$(p, \mu)$                           ▷ proposer $p$, matching $\mu$
        **if** $(\mu(p) = \emptyset \wedge \kappa_p < n)$ **then**
           $q = \ell_p[\kappa_p]$                           ▷ $p$ wants to propose to $q$
           **if** accept$(q, p)$ **then**
               **if** $\mu(q) \neq \emptyset$ **then**                  ▷ break up $q$ if married
                  $r = \mu(q)$; $\mu = \mu \setminus \{\langle q, r \rangle\}$
               $\mu = \mu \cup \{\langle p, q \rangle\}$; $\kappa_q = pr_q(p)$           ▷ match $p$ and $q$
           **else** $\kappa_p = \kappa_p + 1$                     ▷ $q$ rejects $p$

Algorithm 1, POWERBALANCE, applies this principle: it goes through a series of SDA proposal iterations, in each of which the strongest side proposes; if the number of such matchmaking rounds exceeds a *limit* without termination, then POWERBALANCE enforces termination: it tries the COMPROMISE procedure on both sides and chooses the solution that best fits its goal, yielding a stable matching; as cost measure we use either the sex equality cost or the balance cost, introduced in Section 2. Moreover, we can control how fast we reach such a matching by tuning the *limit* of $\mathcal{O}(n)$ proposal rounds that it performs before enforcing the $\mathcal{O}(n^2)$ termination procedure. We contend that a few rounds can bring the two sides at a position of good balance, from which we can enforce termination, with $\mathcal{O}(n^2)$ overall runtime.

## 5 Deferred Local Search

The algorithms discussed in Section 2 can be classified into two types: (i) those that progressively transform an unstable condition to a stable one; and (ii) those that move from one stable matching to a more favorable one by local search. The former are more efficient, while the latter may achieve higher quality in terms of an equity measure, at the price of high worst-case complexity. We propose *Deferred Local Search* (DLS), which first quickly converges to a fair stable matching by POWERBALANCE, and then improves upon this outcome with a few steps of local search in the lattice of all stable matchings. This way, it achieves both efficiency and high quality in terms of equity measures.

**Enhancing Local Search**    The local search procedure in BILS [45] uses *breakmarriage operations* [34], each requiring $\mathcal{O}(n^2)$ time, thus spends $\mathcal{O}(n^3)$ per step to evaluate neighboring solutions produced by breakmarriage on each of $n$ agents. We reduce this cost by exploring the lattice via fine-grained *rotations* [22] rather than bulk breakmarriage operations. We compute all rotations in $\mathcal{O}(n^2)$ [19], and then, in each step, eliminate those *exposed* (i.e., amenable to elimination) in $\mathcal{O}(n^2)$. We also designed an enhanced, rotation-based version of BILS, which we term iBILS.

**Applying Local Search**    Our first *Deferred Local Search* (DLS) proposal, HYBRID, moves ahead from the output of POWERBALANCE, so as to reach a good *neighboring* solution in the lattice of stable matchings via rotation operations. Even in its refined form, BILS starts out from an extreme position in the lattice and proceeds through several $\mathcal{O}(n^2)$ local search steps, amounting to a $\mathcal{O}(n^4)$ worst-case complexity. By contrast, HYBRID starts out from a middle position in the lattice, and performs a controlled number of local search steps, with a $\mathcal{O}(n^2)$ worst-case complexity. Our second DLS proposal, HYBRIDMULTISEARCH (HMS), enforces the termination of POWERBALANCE at different rounds to yield several evenly placed solutions as starting points for local search. Instead of deciding on one of two sides when enforcing termination by COMPROMISE, we use both options as starting points. HMS takes $\mathcal{O}(rn + kmn^2)$ time, where $r$ is the number of POWERBALANCE proposal rounds, $k$ the number of local searches, and $m$ the maximum number of local search steps.

## 6 Experimental Study

We conduct experiments measuring sex-equality cost, balance cost, and runtime. We use synthetic datasets that draw preferences from three distributions: *Uniform(U)*, with preferences created fully *at random*; *Discrete(D)*, where for a *Hot Set* $H \subseteq \mathcal{A}$, if $a_i \in H$ then $a_i \succ_{b_k} a_j, \forall a_j \in (\mathcal{A} - H), \forall b_k \in \mathcal{A}$ ; and *Gaussian(G)*, in which $a_i \succ_{b_k} a_j$ iff $i + X \geq j + Y, \forall b_k \in \mathcal{A}$ for $X, Y = \mathcal{N}(0, 0.4n)$. We also generate asymmetric data set, in which one side follows the *Uniform* model, while the other side follows the *Discrete*; we set the *Hot Set* of *Discrete* distributions to include $40\%$ of the agents. Last, we apply our solution on real data, reported at the end of this section. The algorithms are implemented in Java[4] and tested on an Intel Xeon 2.67GHz CPU with 28GB RAM.

**PowerBalance Parameter Tuning**    POWERBALANCE employs a *limit* parameter, which determines the maximum number of matchmaking proposal rounds it performs before enforcing termination. We experimentally determine a sufficient value for *limit* as a function of dataset size. We generated 100 instances for every size $n$; for each instance, we tested a large number of *limit* values to find out the smallest value that suffices to get the best obtained results on sex-equality ($SEq$). We observed that a sufficient *limit* value grows in a fashion similar to $n\log^2(n)$. In effect, we set the

POWERBALANCE *limit* to $\Theta(n \log^2 n)$, yielding a complexity of $\mathcal{O}(n^2 \log^2 n)$. We set the $k$ and $m$ parameters of HMS to $\Theta(\log n)$, so as to maintain the same asymptotic complexity bound.

**BILS Probability Parameter**    Viet et al. [45] suggest that their bidirectional local search execute random moves with probability $p = 0.05$. We shed light on the impact of $p$, measuring the sex-equality cost of the solution returned by both BILS and iBILS for three different sizes, 2000 instances per size, and a range of $p$ values across distribution types. With iBILS, we observe an improvement in sex equality on Discrete data, peaking at around $p = 0.125$. Contrariwise, BILS did not benefit by randomization, obtaining best results with $p = 0$. This difference is due to that fact that iBILS explores the lattice by rotations, which are smaller steps than the breakmarriages used in BILS.

**Performance Evaluation**    We compare the proposed algorithms against: **APPROX**, the lattice-based approximation algorithm [25]; **POLYMIN**, which finds the solutions minimizing the regret and egalitarian cost and reports the best result; **DACC**, the proposal-based method of [11]; **BILS**, the local-search-based method [46, 45]; and **iBILS**, our own enhancement of BILS. We normalize cost results, dividing by the corresponding best cost the DA algorithm can obtain. Algorithms using local search guide their search using a $SEq$ or $Bal$ cost function; POWERBALANCE selects the best of two outcomes with regard to cost when enforcing termination. DACC [11] does not specify an order of proposals; thus, we employ the proposal strategy of PowerBalance, letting all members of the advantaged side act as proposers in each round. Given our analysis, we set the probability parameter in BILS to 0, in iBILS to 0.125, the *limit* parameter of POWERBALANCE to $\lceil n \log_2^2 n/10 \rceil$, and the parameters in HMS to $k = \lceil 2 \log n \rceil$ and $m = \lceil \log n \rceil$.

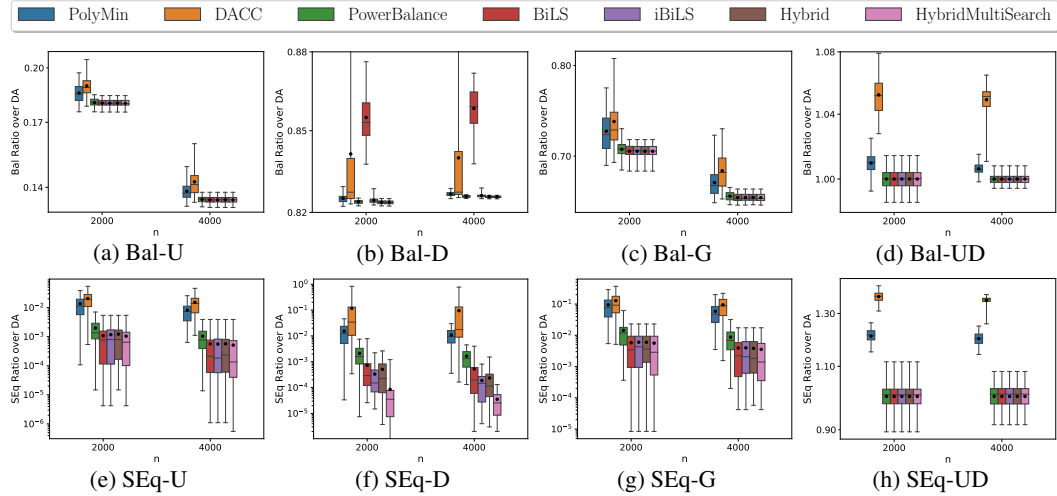

Figure 1: Quality comparison against heuristics

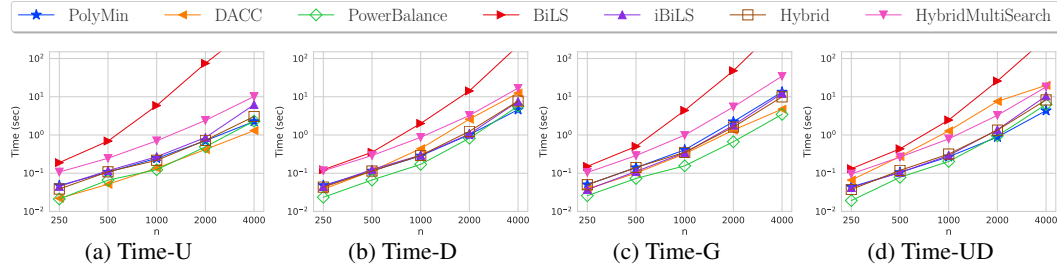

Figure 2: Time comparison against heuristics

**Comparison against other heuristics**    We compare our proposals against state-of-the-art heuristics, on data sizes up to 4,000, with 50 instances per size and distribution and depict cost results with box and whisker plots, with a black dot indicating the mean. On runtime, we plot mean values. Figures 1 and 2 show our results. DACC and POLYMIN perform poorly for both cost metrics. POWERBALANCE is among the fastest, yet falls short cost-wise compared against the local search methods. BILS performs the worst in runtime, while it is also weak in terms of balance and sex

equality cost on Discrete data (Figures 1b, 1f). iBiLS and HYBRID behave similarly, with HYBRID having a slight scalability advantage (Figure 2a). HMS achieves top quality across the board and outperforms others significantly on Discrete data (Figure 1f). Most algorithms detect the same *one-side-biased* solution on UniformDiscrete data (Figures 1d, 1h); due to the innate asymmetry among the two sides, a solution that favors one side over the other achieves good sex equality and balance. Overall, POWERBALANCE is the most scalable, while HMS provides the highest quality.

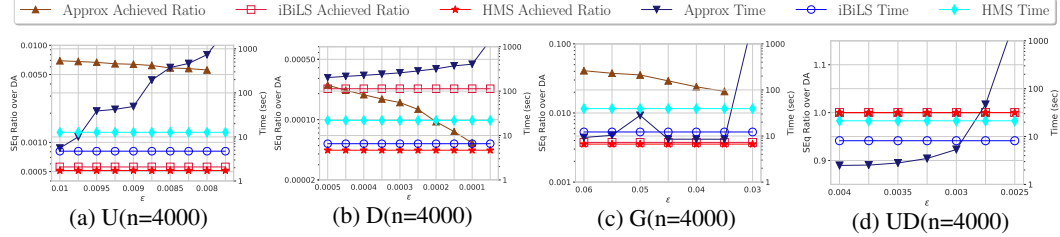

(a) U(n=4000)  (b) D(n=4000)  (c) G(n=4000)  (d) UD(n=4000)

Figure 3: Performance comparison against APPROX

**Comparison against APPROX**   We now test our best methods, iBiLS and HMS, against AP-PROX [24], whose $\varepsilon$ parameter provides a sex-equality approximation guarantee with respect to the best of the two DA outputs. We generate 50 data sets of size 4000 for each distribution, and explore the range of $\varepsilon$ to find values that yield competitive results. Figure 3 presents our results. The axes on the left denote cost ratio (for APPROX, upper-bounded by $\varepsilon$), while those on the right denote runtime. On Uniform and Gauss, iBiLS and HMS significantly outperform APPROX, while the cost ratios they achieve put an overwhelming strain on the latter's runtime (Figures 3a, 3c). On Discrete data, APPROX surpasses the ratios of iBiLS at the cost of a runtime overhead, but does not reach the ratios of HMS within reasonable runtime (Figure 3b). All algorithms find the same solution on UniformDiscrete, while APPROX needs an unnecessarily high runtime with ill-chosen $\varepsilon$ (Figure 3d).

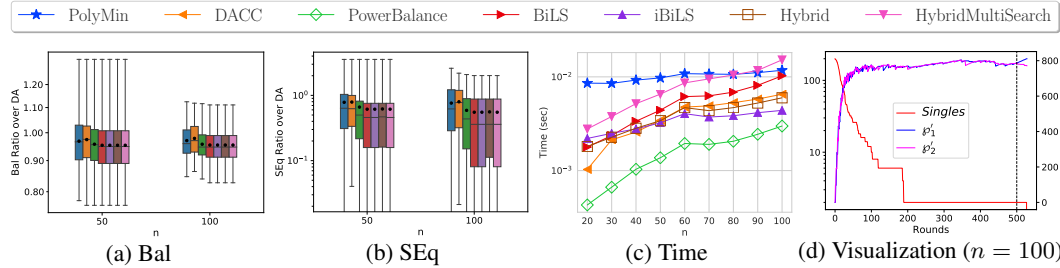

(a) Bal  (b) SEq  (c) Time  (d) Visualization ($n = 100$)

Figure 4: Real Data Experiment

**Application on real data.**   To investigate performance on real data, we extract distributions from the data of an online dating service [10]. The data consists of 17,359,346 anonymous ratings, on the $1 - 10$ scale, of 168,791 profiles made by 135,359 LibimSeTi users, along with gender information. We remove users of unknown gender and those who have not rated the opposite gender, and construct a 2D distribution of the frequency of each pair of ratings $(i, j)$. Drawing from this distribution, we generate data of $n = 100$. We resolve ties using 80% *randomness* and 20% *popularity* (P), i.e., the global ranking of agents by all ratings. We run 50 instances per size, and plot quality and runtime results in Figures 4a, 4b, and 4c. We also visualize, Figure 4d, the process for POWERBALANCE with the instance yielding the *median* sex equality cost; in each iteration, we measure the number of *single* agents and the sum of $\kappa$ index values for the two sides, i.e., $\wp'_m$ and women $\wp'_w$ as defined in Section 2, which dictate which side proposes in the next round. The left-side axis marks the scale of singles, while the right-side axis marks the scale for $\wp'_m$ and $\wp'_w$; the vertical black dashed line shows the round in which POWERBALANCE enters its termination procedure, COMPROMISE.

## 7   Conclusions

We revisited the NP-hard problem of finding a stable matching optimizing an equity measure. We extended the *Deferred Acceptance* algorithm to a two-sided form, *Monotonic Deferred Acceptance*, proposed a simpler variant of its proposal acceptance criterion, *Strongly Deferred Acceptance* (SDA), and amended that with a few selective steps of efficient local search, *Deferred Local Search* (DLS). These are the *first*, to our knowledge, procedures that reach stable matchings of good equity in quadratic time. Our experimental results demonstrate that DLS delivers both efficiency and high equity. In the future, we intend to study the problem under manipulation incentives, as in [36].

## Footnotes

[1]The US resident matching program had used this algorithm since 1952 for junior doctor recruitment [40].

[2]A stable matching *dominates* another when it is strictly preferred by one gender.

[3]Note that at least one partner is always idle, as one must have proposed for the couple to be created.

[4]Code and data are available at `https://github.com/ntzia/stable-marriage`

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
