[Reviews · NeurIPS 2019]

Reviewer 1



The authors propose a novel algorithm for stable matching problem in a two-sided market. This algorithm is a cute extension of the classical DA (Deferred Acceptance) algorithm. The modification includes three important components: - Monotonic Deferred Acceptance - Strongly Deferred Acceptance - local search steps (resulted in Deferred Local Search) The paper is well written and structured. The technical presentation is clear with a good balance between formal notations and informal explanation. The core contribution of the paper is the novel algorithm that achieves O(n^2) complexity (w.r.t. the number of participants in the two-sided market). It would be great to see an evaluation or application of the proposed algorithm in practice (real example). //After rebuttal: thank you for clarification on practical evaluation. Please, make sure to include it in the next version of your paper

Reviewer 2



Originality: This method is a novel extension of existing work. Quality: There are no obvious technical issues with the submission, however, some parts of the writing are unclear and I had to make my own guesses about what authors mean in some places (see Improvements section for suggestions). Clarity: The introduction is clear but the main text is not very clear. In particular, it was hard to understand how the proposed algorithms actually work. Significance: In theory DA produces unfair matchings. In many practical applications there is only a few stable matchings anyway so the unfairness doesn't really matter. This algorithm may be important for cases where indeed there are many stable matchings and we care about fairness.

Reviewer 3



Comments: The problem being addressed is a nice twist on a classical combinatorial optimization problem, and focusing on the fairness aspects of the famously-not-fair stable matchings that pop out of standard DA is a good thing. I found the paper extremely readable, and the authors clearly know the related literature and state of the art in stable matching. Still, some minor comments regarding language/notiation/citations/etc follow: * Was there a reason to use M instead of the standard \mu to represent the matching function/matching? (In fact, \mu is used to denote a matching in Algorithm 1 instead of M.) * It would be good to define a few terms more formally and stick to them. For example, at various times, couples "break up" (L121) or are "abandoned" (L125, L133), there can be "divorcees" (L127) afterward, and so on. * "Idle" in Theorem 3.1 should not be italicized. * I'd be careful about the wording used in the Conclusions, where "finding an equitable stable matching" is NP-hard and then a polytime algorithm for finding "stable matchings of good equity" is proposed. I get what you're saying, but you don't want somebody to write the paper off because of what that might seem to imply. * L220, all agents to an idle state'' not all agents in an idle state'' * n isn't defined in the Algorithm 1 block * The cost of a matching also isn't defined in the algorithm block, not clear if this refers to regret cost, egalitarian cost, balance cost, or some other cost. (And, those costs haven't been discussed for a few pages by the time the reader hits Algorithm 1, so it'd be good to add a sentence near the algorithm block in general.)

[Author Response · NeurIPS 2019]

We thank all reviewers for the time they invested to review this paper and share their insights. In this letter, we respond to all reviewer comments, quoted verbatim, **bold in teal color**; content from the paper is quoted in blue.

**Application of the proposed algorithm in practice; visualization of the matching process; real-world stable marriage data.** We have conducted experiments on real-world data, yet could not include them within page limits. We extract distributions from the data of an online dating service [2]. The data consists of 17,359,346 anonymous ratings, on the $1-10$ scale, of 168,791 profiles made by 135,359 LibimSeTi users, along with gender information. We remove users of unknown gender and those who have not rated the opposite gender, and construct a 2D distribution of the frequency of each pair of ratings $(i, j)$. Drawing from this distribution, we generate data of $n = 100$; the limited scale of ratings does not suffice to generate interesting preference lists at larger size. We resolve ties using 80% *randomness* and 20% *popularity* (P), i.e., the global ranking of agents by all ratings. We run 50 instances per size, and plot quality and runtime results. We visualize the process for POWERBALANCE with the instance that yields the *median* Sex-Equality Cost. In each round, we measure the number of *Single* agents and the sum of $\kappa$ index values from men ($\wp'_m$) and women ($\wp'_w$), which dictate which side proposes in the following round. The left-side axis marks the scale of singles and the right-side axis marks the scale of $\wp'_m$ and women $\wp'_w$. The vertical black dashed line marks the round in which COMPROMISE (the terminating procedure) runs.

| (a) Bal | (b) SEq | (c) Time | (d) Visualization ($n = 100$) |

**Publication of the algorithm in an implemented code (e.g. Java as stated in Line 304).** Please note that a link to code and data is provided in footnote 3 at Line 304, Page 7.

**Pseudocode for the subroutines PROPOSE and COMPROMISE.** The pseudocodes are given below.

```
procedure PROPOSE(p, μ)                    ▷ proposer p, matching μ
    if (μ(p) = ∅ ∧ κ_p < n) then
        q = ℓ_p[κ_p]                        ▷ p wants to propose to q
        if accept(q, p) then
            if μ(q) ≠ ∅ then                ▷ break up q if married
                r = μ(q); μ = μ \ {⟨q, r⟩}
            μ = μ ∪ {⟨p, q⟩}; κ_q = pr_q(p)  ▷ match p and q
        else κ_p = κ_p + 1                  ▷ q rejects p
```

```
function COMPROMISE(C, μ)                   ▷ side C, matching μ
    if (C == M) then F = W
    else F = M
    while (∃x ∈ C : μ(x) = ∅ ∧ κ_x < n) do
        for all x ∈ C do PROPOSE(x, μ)
    while (∃x ∈ F : μ(x) = ∅ ∧ κ_x < n) do
        for all x ∈ F do PROPOSE(x, μ)
    return μ
```

**Relevance to real world matching problems unclear.** As we report in Lines 33–34, citing recent results by Hassidim et al. [3] brought to our attention by Roth [4], the set of possible stable matchings is large in real-world markets [3].

**What does "SEQ ratio over DA" mean?** It means that we normalize cost results, dividing by the corresponding best cost the DA algorithm can obtain; lower cost values are better.

**A characterization of under what cases DA leads to very unfair matchings.** The most challenging cases are those of *symmetric* distributions on two sides, even more so when some choices are universally popular, as in dataset D.

**Minor comments regarding language/notation/citations/etc.** Thank you for these comments, we will heed them.

**"10 million students" in a Chinese admissions market.** We cite this fact as a motivation for research. In most countries the sizes of student admission markets are in the order of a few thousands.

**Why not compare against an IP formulation of the problem?** A mixed-integer linear programming formulation is indeed possible, e.g., minimizing an auxiliary variable $X$ such that $SEq < X$ and $SEq > -X$. To our knowledge, such an IP-based solution has not been attempted to date. Sethuraman et al. [6] consider fairness, yet only in terms of a *median* stable matchng, not in terms of sex-equality cost. Sethuraman opined that an IP-based solution method for sex-equal stable marriage is "unlikely to have polynomial running time" [5]. In a recent study [1], Ágoston et al. were able to solve only a pruning-intensive variant of the stable matching problem using an IP technique on real data; other variants were infeasible for sizes larger than 100; matching 100 students to 20 schools with common quotas took 39,560 seconds (almost 11 hours) [1]. On the other hand, our quadratic-time algorithms surpass or match the quality that APPROX achieves with all carefully tested values of $\varepsilon$ for which a solution exists (Figure 3); thus, they achieve *near-optimal* solutions. We reconfirmed this fact in communication with the authors of APPROX [7].

[1] Kolos Csaba Ágoston, Péter Biró, and Iain McBride. Integer programming methods for special college admissions problems. *Jnl Comb. Opt.*, 32(4):, Nov 2016. 1
[2] Lukas Brozovsky and Vaclav Petricek. Recommender system for online dating service. In *Proceedings of Conference Znalosti 2007*, Ostrava, 2007. VSB. 1
[3] Avinatan Hassidim, Assaf Romm, and Ran I. Shorrer. Need vs. merit: The large core of college admissions markets. https://ssrn.com/abstract=3071873. 1
[4] Alvin E. Roth. Private Communication, 2018. 1
[5] Jay Sethuraman. Private Communication, 2019. 1
[6] Jay Sethuraman, Chung-Piaw Teo, and Liwen Qian. Many-to-one stable matching: Geometry and fairness. *Math. Operations Research*, 31(3):581–596, 2006. 1
[7] Hiroki Yanagisawa and Shuichi Miyazaki. Private Communication, 2019. 1


[Meta-Review · NeurIPS 2019]

The broad consensus of the reviewers was that the authors clearly presented their theoretical results, which are a significant contribution to our understanding of stable matchings, but the algorithm presented would not likely be used in practice, contrary to the pitch made by the authors. Thus, we would urge the authors to reframe the paper to better reflect the strengths of the work.